# Effect of the Application of Ultrasound to Homogenize Milk and the Subsequent Pasteurization by Pulsed Electric Field, High Hydrostatic Pressure, and Microwaves

**DOI:** 10.3390/foods12071457

**Published:** 2023-03-29

**Authors:** Leire Astráin-Redín, Dagbjørn Skipnes, Guillermo Cebrián, Ignacio Álvarez-Lanzarote, Tone Mari Rode

**Affiliations:** 1Departamento de Producción Animal y Ciencia de los Alimentos, Tecnología de los Alimentos, Facultad de Veterinaria, Instituto Agroalimentario de Aragón—IA2—(Universidad de Zaragoza-CITA), 50013 Zaragoza, Spain; 2NOFIMA Norwegian Institute of Food, Fisheries and Aquaculture Research, 4021 Stavanger, Norway

**Keywords:** milk, homogenization, novel technologies, pasteurization

## Abstract

The efficacy of applying ultrasound (US) as a system to homogenize emulsions has been widely demonstrated. However, research has not yet shown whether the effect achieved by homogenizing milk with US is modified by subsequent pasteurization treatments that use new processing technologies such as pulsed electric fields (PEF), microwaves (MW), and high hydrostatic pressure (HPP). The aim of this study was, therefore, to optimize the application of US for milk homogenization and to evaluate the effect of PEF, HPP, and MW pasteurization treatments on the sensorial, rheological, and microbiological properties of milk throughout its shelf life. To homogenize whole milk, a continuous US system (20 kHz, 0.204 kJ/mL, 100%, 40 °C) was used, and different ultrasonic intensities (0.25, 0.5, and 1.0 kJ/mL) were evaluated. The optimal ultrasonic treatment was selected on the basis of fat globule size distribution and pasteurization treatments by MW (5800 W, 1.8 L/min), PEF (120 kJ/kg, 20 kV/cm) and HPP (600 MPa, 2 min, 10 °C) was applied. The ultrasound intensity that achieved the highest reduction in fat globule size (0.22 ± 0.02 µm) and the most homogeneous distribution was 1.0 kJ/mL. Fat globule size was smaller than in commercial milk (82% of volume < 0.5 µm for US milk versus 97% of volume < 1.2 µm for commercial milk). That size was maintained after the application of the different pasteurization treatments, and the resulting milk had better emulsion stability than commercial milk. After 28 days of storage, no differences in viscosity (4.4–4.9 mPa s) were observed. HPP pasteurization had the greatest impact on color, leading to higher yellowness values than commercial milk. Microbial counts did not vary significantly after 28 days of storage, with counts below 10^2^ CFU/mL for samples incubated at 15 °C and at 37 °C. In summary, the homogenization of milk obtained by US was not affected by subsequent pasteurization processes, regardless of the technology applied (MW, PEF, or HPP). Further research is needed to evaluate these procedures’ effect on milk’s nutritional and functional properties.

## 1. Introduction

Consumer demand for natural, minimally processed products has been steadily increasing over the last few years. Minimal processing technologies are thus being increasingly introduced at different food processing stages due to benefits including low impact on the product’s sensory and nutritional quality, lower energy costs, and shorter processing times. Four of these minimal processing technologies are pulsed electric field (PEF), ultrasound (US), high-pressure processing (HPP), and microwaves (MW). A considerable amount of research has been conducted on these technologies [1,2,3,4,5,6,7,8]; however, the same technologies can be applied to different food processes with varying effects. It is thus important to ascertain the role of a series of processing parameters in the application of these technologies. The dairy industry is one of the sectors where their application has been more thoroughly investigated, with the aim of improving the bioactivity, functional properties, and sensory quality of milk [9].

Ultrasound (US) consists of the propagation of sound waves greater than 20 kHz through air, liquid or solid media, generating cycles of high and low pressure: in liquid media, it leads to the cavitation of microbubbles. As a secondary effect, US forms microcurrents and microjets that facilitate mass and energy transfer processes [10,11]. Studies have been carried out on a series of different applications of high-power US in the dairy industry: authors have reported the improvement of lactose crystallization in milk, an increase in its bioactive activity, improved rheological properties, and a reduction of fat globule size [12,13,14,15,16,17,18]. The latter achievement is of great importance in view of the fact that the traditional homogenization process is applied to all commercial milk with the aim of stabilizing the emulsion by reducing the size of its fat globules, thereby avoiding oxidation and phase separation [19]. When US is applied for this purpose, the sound wave propagation initially leads to the dispersion of the fat droplets in the aqueous phase; cavitation then promotes the breakage of the fat droplets [20]. Emulsions obtained by US have a smaller particle size and higher homogeneity than those obtained by conventional methods; less energy is thus required for emulsion formation, and it is not necessary to add a surfactant to improve stability [21]. However, the application of high ultrasonic energies can lead to the opposite effect: namely, the re-coalescence of the fat globules into larger globules [22]. Moreover, the application of high intensities has a negative effect on the milk’s sensory and nutritional qualities due to the increase in temperature. It is, therefore, necessary to adjust US treatment conditions to achieve a reduction in the size of the fat globules without increasing the product’s temperature [23]. 

The processing stage of commercial milk in which the product suffers the greatest thermal impact is pasteurization, which takes place after homogenization. Conventional microbial inactivation treatments are based on heat, i.e., high-temperature short-time (HTST) or low-temperature long-time (LTLT) pasteurization. New alternatives are being investigated: either thermal technologies that allow for rapid volumetric heating (ohmic heating, microwave (MW), radio frequency) or non-thermal technologies (PEF, HPP, cold plasma, ultraviolet light) that preserve the characteristics of fresh milk. PEF consists of the application of electric fields of high intensity (>0.1 kV/cm) and short duration (from milliseconds to microseconds) to a product placed between two electrodes [24]. The application of pasteurization treatments by PEF is feasible in liquid foods such as fruit juices, vegetable juices, milk, and liquid egg, provided that further “hurdle” technologies such as mild heat or antimicrobials are likewise applied [25,26,27]. Since the applied temperature is not very high, the impact on nutritional quality is minimal [28]. 

Another non-thermal food preservation technology is HPP, which is usually applied after packaging to reduce the risk of post-processing recontamination. Several authors have investigated HPP as an alternative to conventional pasteurization of milk [29]. For instance, Liu et al. [30] compared HPP (600 MPa/5 min) with HTST treatment (72 °C/15 s) and found no significant differences in terms of microbial counts, gastricin-vitro protein digestion, or volatile compounds. 

Microwaves (MW) used for the pasteurization of liquid foods result in volumetric heating by dielectric transfer of energy [2]. The major advantage of microwaves over conventional thermal systems is that they attain high temperatures in a very short time while preserving the product’s sensory and nutritional quality [31]. Several reviews have been conducted on the subject of applying MW to dairy products [2,32], but none of them have taken into consideration its effect on the distribution of fat globule size in milk. 

In view of their promising potential for the dairy industry, all these technologies have been extensively investigated—applied either individually or in succession [33]—for purposes of improving the preservation of milk and milk-derived products. However, most of those studies have only focused on evaluating the treatments’ microbial lethality; moreover, none of them has evaluated if the impact of a novel technology on a certain characteristic is maintained or modified by the successive application of the others. More specifically, the effect of homogenizing milk by US and then pasteurizing the same milk through novel technologies such as PEF, HPP, and MW has not yet been investigated. 

The objective of this study was, therefore, to evaluate a new approach consisting of the application of novel technologies to two successive stages of milk processing. First, the application of US was optimized for milk homogenization, and subsequently, the effect of a pasteurization treatment (PEF, HPP, or MW) on the milk’s sensorial, rheological, and microbiological properties throughout its shelf life.

## 2. Material and Methods

### 2.1. Raw Materials and Experimental Design

Studies were conducted on whole raw milk (4.5% fat) obtained from the Norwegian dairy company Tine (Sola, Norway), which also supplied us with commercial milk stemming from the same batch of raw milk used for our experiments. The dairy company processed the commercial milk in a continuous flow process, including thermization to 68 °C during high-pressure homogenization (TetraPak, Pully, Switzerland) at 40 bars, followed by pasteurization at 72 °C for 15 s. The raw milk was stored at 4 °C for a maximum of 24 h before use. Figure 1 shows an overview of the stages and technologies applied in this investigation. Two replicates from two different batches on separate days were used. 

### 2.2. Homogenization by Continuous US System

The continuous US system used consisted of a transducer and a 1000 W ultrasonic generator (nominal power) with a frequency of 20 kHz (UIP1000hdT generator with the BS4d22 sonotrode, Hielscher Ultrasonics, Berlin). Amplitude was adjustable in the range of 20 to 100%, and the system was capable of processing 0.5 to 4.0 L/min. In addition, the treatment chamber (177 mL volume) had a cooling jacket to control temperature during US processing.

Prior to the US treatments, the raw milk was preheated to 40 °C to favor the homogenization process; this temperature was maintained during the US treatment. The working flow rate was 0.8 L/min, applying an amplitude of 100% (520 W). The treatment parameters were calculated according to energy density, as shown in Equation (1) [22]. The energy densities evaluated were 0.25 kJ/mL (1.4 min), 0.50 kJ/mL (2.8 min) and 1 kJ/mL (5.7 min).
(1)Energy density kJmL=Nominal power kW×Processing time (s)Volume (mL)

Furthermore, the actual ultrasonic energy delivered to the sample was calculated using a calorimetry technique similar to that performed by Both et al. [34]. A power density of 106 W was determined, from which actual energy densities of 0.051, 0.102 and 0.204 kJ/mL were derived for nominal energies of 0.25, 0.50 and 1 kJ/mL, respectively. 

### 2.3. Pasteurization Treatments

#### 2.3.1. HPP System

The HPP equipment used in this experiment was a lab-scale high-pressure QFP 2L-700 unit (Avure Technologies Inc., Columbus, OH, USA). Samples were pressurized at 600 MPa for 2 min at 8–9 °C. Come-up time was approximately 110 s (up to 600 MPa), whereas pressure release was immediate. The duration of treatment did not include the come-up time. Before processing, 40 mL samples of US-homogenized milk were vacuum-packed (99.5%) in sous-vide bags and chilled in cold water up to 9–10 °C. After HPP, the samples were cooled on ice and stored for up to 28 days at 4 °C until further analysis. The experiments were performed twice on different days.

#### 2.3.2. PEF System

For PEF pasteurization treatment, a PEF Pilot Dual System (ELEA Technology GmbH, Quakenbrück, Germany) in continuous mode was used. The device works at a maximum output voltage of 24 kV and a maximum current of 1000 A, with a frequency of up to 500 Hz. The treatment chamber was a collinear chamber with a 1 cm gap. 

The US-homogenized milk at 40 °C was prompted at 2.8 L/min into the PEF equipment to receive treatment of 20 kV/cm, 120 kJ/kg, treatment time of 60 µs, and a final temperature of 69 °C. Immediately after exiting the treatment chamber, it was cooled down to 20 °C by means of a cooling system coupled to the PEF equipment. The milk samples were collected and distributed into sterile bottles under pressure in a safety cabinet in a clean room and stored for 28 days at 4 °C. The experiments were performed twice on different days. 

#### 2.3.3. MW System

A pilot plant MW autoclave (Gigatherm AG, Flawil, Switzerland), as described by Altin et al. [35] and Rosnes & Skipnes [36], was employed, operating at 2450 MHz and set values of 5.8 kW and 1.0 bar pressure set above that of the atmosphere. In order to apply a continuous treatment, a tube made of a thermoplastic elastomer (of the same type as High Temp ultra-barrier, EJ Beverage, Denver, CO, USA) was introduced into the autoclave as a spiral in such a way that the residence time of each sample inside the autoclave was 33 s. The equipment had three fiber optic probes connected along the length of the tube (start, middle, and end) connected to the system’s controller. The US-homogenized milk entered the system at 40 °C at a flow rate of 1.8 L/min. The power applied was 5800 W in order to reach a final temperature of 76 °C. Immediately after heating, the milk was chilled to 20 °C by means of a tube-in-tube cooling system connected to the outlet of the autoclave. Milk samples were collected as described for PEF (Section 2.3.2) in an aseptic procedure and stored for 28 days at 4 °C. The experiments were performed twice on different days.

### 2.4. Droplet Size Distribution

To determine the size of the fat globules in the milk samples after homogenization by US, the Zetasizer Nano ZSP equipment (Malvern Instruments, Malvern, Worcestershire, UK) was used. In order to analyze only the fat globules and to disrupt the casein micelles, 50 mM ethylenediaminetetraacetic acid (EDTA)/NaOH pH 7.0 buffer (Thermo Fisher Scientific, Waltham, MA; United States) was added in a 1:1 ratio [37]. Analyses were carried out at a temperature of 20 °C, and values of 1.462 and 1.333 were used for the refractive index of milk and water, respectively [38]. The z-average value, which derives the mean value from the intensity distribution, was used to compare the different ultrasonic treatments. Analyses were carried out in triplicate.

### 2.5. Zeta (ζ)-Potential Measurements

The ζ-potential was determined by electrophoretic light backscattering using Zetasizer Nano ZSP equipment (Malvern Instruments, Malvern, Worcestershire, UK). Milk samples were prepared in the same way as in Section 2.4. Analyses were carried out before and after homogenization, after pasteurization, and during the milk storage period (up to 28 days). Each analysis was performed in triplicate.

### 2.6. Color

VeriVide’s DigiEye system (VeriVide Ltd., Leicester, UK) was used to determine the color of the milk samples before and after homogenization, after pasteurization, and during storage (28 days). This system was equipped with a DSLR camera (Nikon D90, Tokyo, Japan), and it captured an image of 4288 × 2848 pixels with a resolution of 96 dpi. Before each measurement, the camera was white-balanced and calibrated with the color chart provided with the equipment. The *L**, *a**, and *b** components were recorded at a D65 standard illuminant. Each sample specimen was photographed in a light cabinet on a white background. Pictures were analyzed with DigiPix software (VeriVide Ltd., Leicester, UK) according to the CIELAB color scale. Data were obtained from 10 measurements of each sample, and each analysis was performed in triplicate. To determine color, we used the *L* a* b** color space. Moreover, total color differences (ΔE) of the samples before (raw milk) and after processing and storage were calculated as indicated in Equation (2):(2)$$\Delta E=\sqrt{\left[{\left(L_{0}^{*}-L^{*}\right)}^{2}+{\left(a_{0}^{*}-a^{*}\right)}^{2}+{\left(b_{0}^{*}-b^{*}\right)}^{2}\right]}$$
where $L_{0}^{*}$, $a_{0}^{*}$, and $b_{0}^{*}$ were the values for the untreated sample (control).

### 2.7. Rheological Behavior

Milk viscosity was determined using a hybrid rheometer (Discovery HR-2,TA Instruments, Newcastle, UK) and cone-plate geometry (40 mm, 2°) at 22 °C. Approximately 1 mL of the milk sample was loaded onto the cross-hatched Peltier plate, and flow curves (shear stress vs. shear rate) were determined at increasing shear rate: 0–1200 s^−1^ in 2 min, up and down the flow curves [39]. TRIOS version 5.6.0.87 software. (T.A. Instruments, New Castle, DE, USA) was used for the analysis of results. Results were fitted to the Power Law Model [40], described by the following equation:(3)$$\sigma =K{\dot{\gamma }}^{n}$$
where σ is the shear stress (Pa), $\dot{\gamma }$ is the shear rate (s^−1^), *K* is the consistency coefficient (Pa s^n^), and *n* is the flow behavior index. The flow behavior index (*n*) for shear-thinning liquids ranges from 0 to 1. Moreover, the apparent viscosity (Pa s) was calculated by dividing shear stress (Pa) by shear rate (s^−1^). 

Analyses were carried out in triplicate before and after homogenization, after the pasteurization treatments, and during storage (up to 28 days). 

### 2.8. Microbial Growth during Storage

After applying the pasteurization treatments, microbial growth was studied during storage (4 °C) on days 7, 14, and 21. For microbial analyses, triplicate samples of milk were diluted 1:10 in peptone water (Merck) and vortexed. Quantification of the total number of aerobic bacteria in the milk (colony-forming units CFU/mL) was performed by surface plating on plate count agar (PCA, Oxoid), mainly using a mechanical spiral plater (Eddy Jet, IUL Instruments, Barcelona, Spain). Plates were incubated for 24 and 48 h at 15 °C and at 37 °C. 

### 2.9. Statistical Analyses

Flow curves were fitted to the Power Law Model by applying the least-squares fitting criterion with Microsoft Excel software (version 2017, Microsoft Corporation, Washington, WA, USA). The goodness of fit was determined by calculating the coefficient of determination (R^2^). GraphPad PRISM 9.0 software (GraphPad, San Diego, CA, USA) was used for statistical analyses (1-way ANOVA with Tukey post-test), and significant differences were calculated using *p* = 0.05. Error bars in the figures correspond to the standard deviation of the means.

## 3. Results and Discussion

### 3.1. Fat Globule Size Distribution

Fat globule size was first evaluated by observing the application of three different ultrasonic energy densities (0.25, 0.5, and 1 kJ/mL). The fat globule size of the raw milk samples exceeded the equipment’s upper detection limit. Figure 2 shows the average size (z-average) of the three applied ultrasonic intensities. It can be seen that the higher the energy, the smaller the fat globule size, all of them lying below the average value of commercial milk. The average size of the fat globules in commercial milk was 0.44 ± 0.04 µm, whereas when US was applied, it decreased to 0.32 ± 0.04, 0.28 ± 0.02, and 0.22 ± 0.02 µm for 0.25, 0.50, and 1.00 kJ/mL, respectively. 

However, the fat globules’ overall size distribution varied according to US treatment intensity as well as under the effect of the novel technologies we subsequently applied. Figure 3A shows results after US homogenization at different US intensities, including the commercial milk batch for comparison, and Figure 3B shows results following the application of the subsequent treatments (PEF, HHP, and MW). It should be noted that the milk is considered homogenized when the size of the fat globule is reduced to 1 µm [41]. For the commercial milk used in this experiment, 85.3% of the fat globules had a size of 1.7 µm, and 14.7% had a size of 0.34 µm.

In the case of samples only homogenized by US (Figure 3A), globule size distribution was not homogeneous in all cases. At 0.25 and 0.50 kJ/mL, a second peak was observed with a mean size of 3.63 ± 1.22 and 4.85 ± 0.70 µm, whereas distribution at 1.00 kJ/mL was more homogeneous: 95% of the globules had a mean size of 0.24 µm. These results indicated that US helped to reduce the overall fat globule size as compared to commercial milk, especially when 1 kJ/mL US was applied.

Most previous studies in this field have used the D criterion [3,4] to analyze results (the volume-weighted mean diameter), whereas in our study, due to our equipment’s characteristics, we used the *z*-average value (the mean value derived from the intensity distribution). Bearing this in mind, previous studies have shown that the application of US can reduce fat globule size to less than 1 µm [42]. Bermúdez–Aguirre et al. [43] applied a US treatment of 1.4 kJ/mL, 24 kHz, 63 °C for 30 min on whole milk and observed that the surface of the fat globule was completely roughened, and the size of the fat globules was less than 1 µm. However, the application of high ultrasonic intensities has also been shown to cause the opposite effect, i.e., an aggregation of fat globules due to the phenomenon of re-coalescence. Scudino et al. [22] observed that large droplets (>10 µm) appeared when US (475 W, 19 kHz) was applied at energy densities of 7 kJ/mL. This may be due to the fact that the application of US disrupts the fat globule membrane; other components, such as casein particles and serum proteins, can thus be adsorbed at the interface, thereby creating new bridges among the fat globules [37]. 

Based on these results, we selected a US treatment of 1 kJ/mL to homogenize the raw milk for further studies, as well as the milk pasteurized by MW, HPP, and PEF treatments (Figure 3B). As observed, MW, HPP, and PEF did not increase the size of fat globules obtained through US homogenization. Furthermore, no significant differences (*p* = 0.05) among z-average values were found between samples immediately after US-homogenization (0.22 ± 0.02 µm) and samples that had been homogenized and subsequently pasteurized (0.21 ± 0.04, 0.21 ± 0.05, and 0.20 ± 0.05 µm for MW, HPP, and PEF samples, respectively). 

These results indicated that the size and distribution of the fat globules achieved after a US treatment of 1 kJ/mL were maintained after the pasteurization treatments, independently of the applied technology (PEF, MW or HPP). To the best of our knowledge, there are no previous studies in the literature on milk or other foods that were homogenized with US and subsequently pasteurized or sterilized with new technologies. Nevertheless, some research has been carried out using these technologies individually. In a research conducted by Sharma et al. [44], they studied the effect of PEF treatment on the milk fat globule membrane. The results obtained showed that the application of a PEF treatment at 20 kV/cm for 34 µs did not significantly affect the size of the fat globule. However, applying a heating pre-treatment (55 °C/24 s) and higher electric fields (26 kV/cm) decreased the fat globule size, increased the surface area and increased the adsorption of caseins and whey proteins on the fat globule membrane. For HPP treatment, it has been found that the size and composition of the fat globule membrane could be altered [45]. For instance, Ye et al. [46] applied a 100–800 MPa treatment in whole milk and observed an increased association of beta-lactoglobulin to the fat globule membrane via sulphhydryl disulfide interactions and at higher pressures (500–700 MPa) small amounts of casein bound to the fat membrane. In the case of MW treatment, it has been found that non-thermal effects mainly affect milk proteins by altering their quaternary and tertiary structures [47], i.e., highly intense MW treatment can lead to casein unfolding [48] and whey protein aggregation [49].

### 3.2. Stability of Fat Globules

Table 1 reports the ζ-potential values of the fat globule emulsion as they evolved along the shelf life of milk samples subjected to different treatments. Raw milk, which was the most unstable one, showed the highest values of −11.4 at day 0 and −12.1 after 7 days of storage at 4 °C. These results are in line with those reported in the literature [50,51]. 

The same lowercase letter in the same row indicates that there are no significant differences (*p* = 0.05) for the same technology over the shelf life. The same capital letter in the same column indicates that, for the same day of shelf life, there are no significant differences (*p* = 0.05) between the applied technologies.

Homogenization and the consequent reduction of fat globule size made the emulsion more stable, displaying significant differences (*p* = 0.05) between ζ-potential values of US-homogenized milk (−17.5 ± 1.2 mV) and conventionally homogenized milk (−14.6 ± 1.0 mV) at day 0 of shelf life. In both cases, these values remained stable throughout storage. 

For the milk pasteurized by PEF, MW, and HPP, ζ-potential values at day 0 were in the same range (−18.5 ± 1.3, −20.1 ± 1.1 and −19.7 ± 1.2, respectively) but still significantly different (*p* = 0.05) in comparison with samples that had only been sonicated. However, these differences were smoothed out during shelf life. These findings confirmed the above-mentioned results regarding improved stability (in terms of size distribution) of fat globules due to US treatment (Figure 3); they also confirm that this stability is maintained not only just after the pasteurization treatment but all along subsequent shelf life: in this case, up to 28 days. 

Pegu & Arya [52] studied the effect of several different US treatments on milk stability. They observed values of −23.4 mV when applying US at 200 W and 1 kJ/mL, whereas the application of higher powers (300 and 400 W) and intensities (1.44–3.84 kJ/mL) did not modify the values, which remained at −23.6 mV for 300 W and 1.44 kJ/mL and at −23.2 mV for 400 W and 3.84 kJ/mL. In the case of PEF, it has been observed that the application of electric fields of 20 kV/cm and 26 kV/cm for 34 µs with preheating to 55 °C significantly reduces the ζ-potential of milk from −13.7 (raw milk) to −14.75 and −14.8 mV, respectively [44]. In our study, no such reduction of the ζ-potential as a result of applying the PEF treatment was observed: this may be due to the fact that we applied the PEF treatment after homogenization; emulsion stability was thus already high, so the effect of PEF could be masked by the US-homogenization treatment. On the other hand, the application of HPP can promote flocculation, which, in turn, can lead to higher mean size values and higher ζ-potential values due to the denaturation of whey proteins that promote interaction between proteins and fat globule membranes [53]. Our results do not seem to indicate that flocculation was taking place, perhaps because the HPP treatment was less intense than in previous studies, where 400 MPa/15–20 min or 500 MPa/10 min were applied [54], or because the stabilizing effect of US-homogenization may have prevented the negative effect of HHP on the emulsion’s stability. 

### 3.3. Color Analysis

Ultrasonic homogenization resulted in whiter milk featuring *L** values of 99.62 ± 0.05, whereas raw milk displayed *L** values of 95.72 ± 0.30. The homogenization process makes milk whiter as it reduces the size of dispersed fat globules, which increases light scattering [55]. Figure 4 shows the total color difference (Δ*E*) during the shelf life of the milk samples compared to commercial milk on each day of measurement (as a reminder, the milk samples were homogenized by US and subsequently pasteurized by PEF, HPP, and MW, whereas the commercial milk was conventionally homogenized and heat-pasteurized). 

The figure shows that the processes under study (involving the application of novel technologies for purposes of homogenization and pasteurization) did not have a notable impact on color in comparison with the values of commercial milk: Δ*E* values remained below 2 throughout shelf life (a value considered undetectable by the human eye [56]). Moreover, by day 28, the Δ*E* value of HPP-pasteurized milk decreased, i.e., it was closer to that of commercial milk. This was due to the fact that both commercial milk and US + HPP milk displayed a higher yellowness value by the end of their shelf life: 17.02 ± 0.13 and 18.67 ± 0.14, respectively. This yellowing effect of HPP has also been reported by other authors [53,57]. This occurs when the HPP treatment is quite intense and results in the disruption of the casein micelles, which would make the milk become translucent, exposing the whey [57].

In the case of PEF technology, McAuley et al. [58] likewise observed no statistically significant differences (*p* < 0.001) in *L*, a** and *b** values when comparing a PEF pasteurization treatment (30 kV/cm, 22 µs and outlet temperatures of 53 and 63 °C) with conventional heat treatment (72 °C/15 s). The same tendency was reported by Clare et al. [59] for microwave technology. These results are in agreement with ours, which show that MW and PEF technologies had the least impact on the color of milk homogenized by US. 

### 3.4. Rheological Behavior

The flow curves obtained from the samples were fitted to the Power Law model: the consistency coefficient value (*K*) and the flow behavior index (*n*) are given in Table 2.

After the US homogenization treatment, both values were similar to those of commercial milk, showing no significant differences (*p* = 0.05). In addition, once pasteurization treatments were applied, no substantial changes appeared in the behavior index (*n*) immediately after treatment or during shelf life. After the US homogenization treatment, the consistency index displayed significantly lower values compared to commercial milk, and those values remained similar when HPP and PEF were applied. The K-index was slightly higher after MW treatment, although not significantly different from that of HPP and PEF milk. By the end of the 28-day storage period, no significant differences (*p* = 0.05) were observed among all pasteurized samples.

Figure 5 shows the apparent viscosity of the samples throughout storage: no statistically significant differences among treatments were observed. 

The average viscosity (measured at 20 °C) was 3.72 ± 0.18 mPa s at the beginning of the storage period, and it rose to 4.73 ± 0.29 mPa s (measured at 20 °C) after 28 days of storage. Certain studies indicate that the application of HPP to milk leads to an increase in viscosity due to aggregation of casein, although this effect depends on treatment intensity [45]. Michalac et al. [60] applied a PEF treatment of 35 kV/cm, 64 pulses, and a final temperature of 52 °C to milk samples and found no significant differences (*p* = 0.05) in viscosity compared to milk thermally pasteurized at 73 °C/30 s. However, Xiang et al. [61] observed an increase in the viscosity of skimmed reconstituted milk after applying 12–20 kV/cm. Although under the conditions studied in this article, no impact on milk viscosity was observed, there are studies in the literature that have found an influence. Therefore, further studies evaluating a range of conditions will be necessary to determine the impact of these novel technologies on the viscosity of treated milk. 

### 3.5. Microbiological Shelf Life

Figure 6 shows the total bacterial counts in milk obtained after homogenization and application of MW, PEF, and HPP treatments. Bacterial counts for commercial milk are likewise included. Initial counts of raw milk were 3.1 × 10^3^–2.2 × 10^4^ CFU/mL.

In all cases, agar plates were incubated at 37 °C (mesophilic bacteria; Figure 6A) and at 15 °C (psychrophilic bacteria; Figure 6B). The counts for aerobic mesophilic bacteria (37 °C) immediately after the treatments (Day 0) were 2.23, 1.98 and 2.10 Log CFU mL^−1^ for the PEF, HPP, and commercial milk, respectively. In the case of the MW treatment, counts were below the detection level (LoD = 1.4 Log CFU mL^−1^). Regarding psychrophiles, and except for HPP-treated milk (in which a mean value of 2.50 Log CFU mL^−1^ was determined), milk treated by all the other technologies—including commercial milk—had values that lay below the detection level. Since the shelf life of pasteurized milk stored under refrigeration is usually 12–21 days, we evaluated the microbiological evolution of the milk samples over a 21-day period. For aerobic mesophiles as well as for psychrophiles, the microbial counts in all the milk samples remained below the detection level, except for commercial milk, which on Day 21 had a count of 1.62 Log CFU mL^−1^ at 37 °C. In summary, microbiological results obtained for the milk samples treated with MW, HHP, and PEF—after US homogenization—were comparable with those of the commercial milk we tested.

Several studies have demonstrated the capacity of MW for the inactivation of microorganisms such as *Escherichia coli*, *Salmonella* sp., *Staphylococcus aureus*, and *Pseudomonas aeruginosa* in dairy products [32]. Thum et al. [62] studied the application of MW treatment (2455 MHz, 75 °C, ramp time of 4.10 min, and holding time of 20 s) for the inactivation of *E. coli* and *S. aureus* in bovine milk; they achieved a five-log reduction for both types of bacteria. However, MW has a non-thermal effect, leading to rapid denaturation and aggregation of whey proteins. Nonetheless, the use of MW allows for rapid volumetric heating, leading to an improved sensory quality of the final product by minimizing browning and off-flavor production [47]. 

Similarly, a number of studies have been conducted on the inactivation of bacteria by HPP [3,63]. Stratakos et al. [53] conducted a shelf-life study of milk samples to which an HPP treatment of 600 MPa/3 min was applied, achieving a microbial reduction of 3.95 Log CFU mL^−1^. In addition, the HPP treatment extended the milk’s shelf life from 7 days (thermal pasteurization in a water bath at 72 °C/5 min) to 14 days by considerably reducing the levels of total viable counts (TVC), *Enterobacteriaceae*, lactic acid bacteria (LAB), and *Pseudomonas* spp. The authors concluded that HPP could be regarded as a viable alternative to heat for milk pasteurization, as was also established by Liu et al. [30] and Yang et al. [64].

On the other hand, the application of PEF alone does not achieve microbial inactivation levels equivalent to those of pasteurization; it is thus necessary to combine it with other technologies, such as heat [65]. The advantage of PEF, however, is that the food’s sensory quality is only minimally affected because the time the product has been taken up to mid-range temperatures is only in the order of micro- to milliseconds. In order for these combined technologies to serve as an alternative to heat pasteurization—i.e., for them to achieve optimal microbial inactivation—it is important to apply the correct combination of PEF and heat [66]. We thus additionally verified that the treatment we applied in this investigation (20 kV/cm, 120 kJ/kg, and a final temperature of 69 °C) led to the inactivation of 5-Log of *Listeria monocytogenes* STCC 5672 (Appendix A). Microbial inactivation by PEF in dairy foods has already been extensively reviewed by Buckow et al. [67]. Sepulveda et al. [68] applied a PEF treatment of five exponential decay pulses, 2.3 µs wide and 35 kV/cm, to milk samples, reaching a maximum temperature of 65 °C. Samples stored at 4 °C took up to 24 days to exceed the established limit of 4.3 Log CFU mL^−1^ for total bacteria. McAuley et al. [58] applied square pulses of 1.5 µs at an electric field of 30 kV/cm and an outlet temperature of 63 °C to raw milk that was subsequently stored at 8 °C. According to their results, after 8 days, there was no significant difference (*p* < 0.001) between the PEF treatment and the heat treatment at 72 °C/15 s, with bacteria counts of 3.8–4.2 Log CFU mL^−1^ for both technologies. 

## 4. Conclusions

US is an effective technology for the reduction of fat globule size, obtaining even better results than the conventional homogenization procedure widely used in the dairy industry. Under our experimental conditions, the best results were obtained with an intensity of 1 kJ/mL at 20 kHz. The greater degree of reduction in fat globule size observed after US homogenization treatments, as compared to commercial milk, also resulted in a more stable emulsion as compared to the latter; the same degree of stability was maintained along storage time. Fat globule size and emulsion stability achieved by US homogenization were likewise maintained after the subsequent application of pasteurization treatments by means of the three technologies under study: PEF, MW, and HPP. 

As ascertained by the ΔE parameter, the milk’s color and its rheological properties were not significantly affected after the application of the PEF, MW, and HPP pasteurization treatments, although the HPP treatment led to a significant increase in yellowness values. Similarly, all the technologies we tested were effective in terms of microbial inactivation. Microbial counts determined along storage time under refrigeration conditions were comparable to those of commercial milk.

In summary: the most relevant finding of this study is that the application of pasteurization treatments by means of novel technologies such as PEF, MW, and HPP did not affect the size distribution of fat globules present in milk homogenized by ultrasound. However, further studies are needed to evaluate the impact of this combination of novel technologies on the nutritional and functional properties of milk.

## Figures and Tables

**Figure 1 foods-12-01457-f001:**
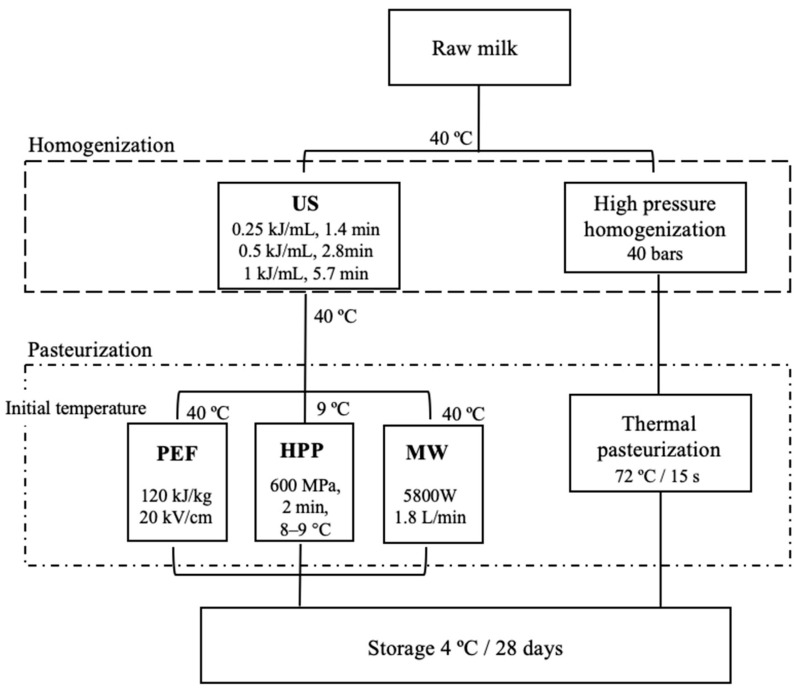
Scheme of the technologies used at each stage of the processing of the milk samples.

**Figure 2 foods-12-01457-f002:**
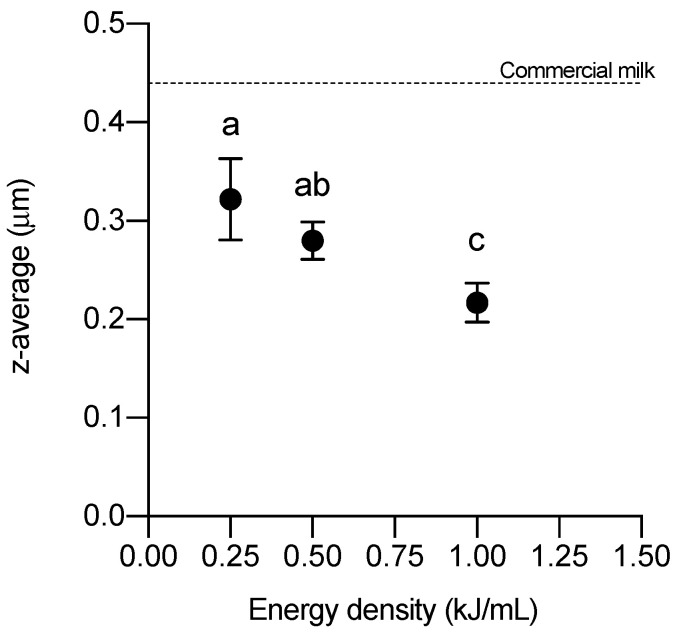
Average size (*z*-average) of milk fat globules after the application of three different ultrasonic intensities. The same lowercase letter indicates that there are no significant differences (*p* = 0.05).

**Figure 3 foods-12-01457-f003:**
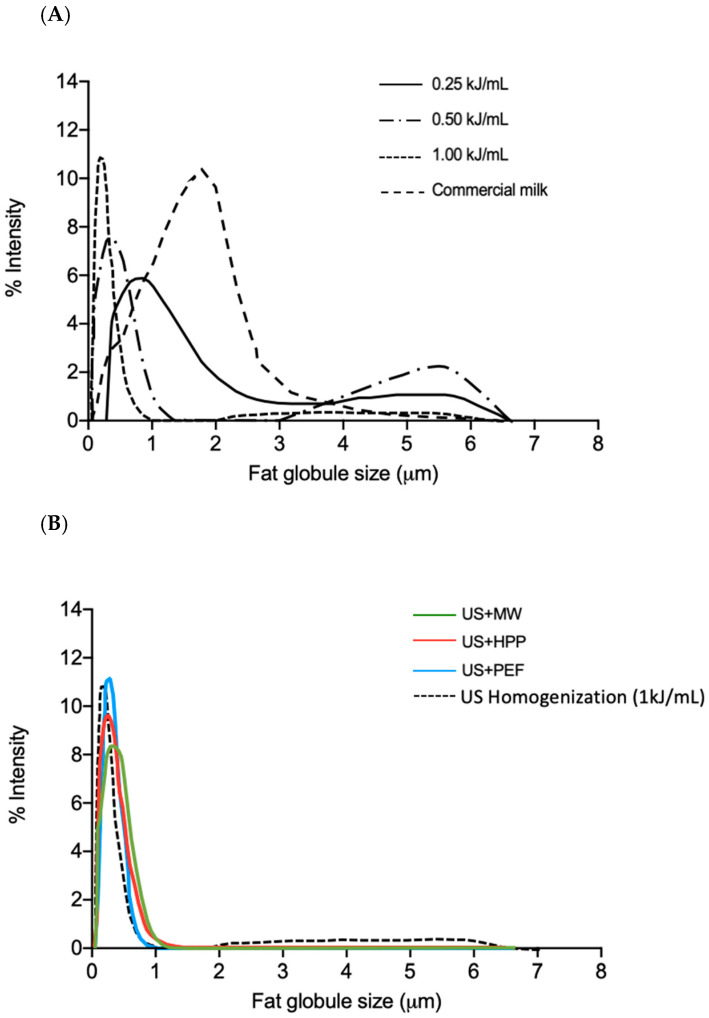
Size distribution of milk fat globules: (**A**) after US homogenization at different energies; (**B**) after US homogenization at 1 kJ/mL (US), followed by MW (US + MW), HPP (US + HPP), and PEF (US + PEF) pasteurization.

**Figure 4 foods-12-01457-f004:**
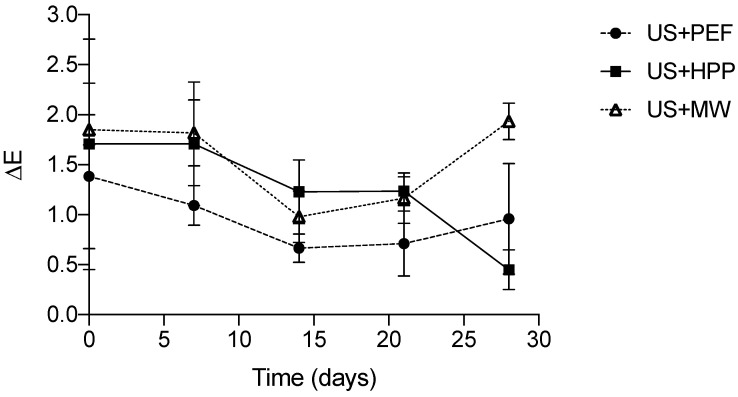
Determinations of the total color difference (Δ*E*) of milk samples during shelf life compared to commercial milk on each day of measurement.

**Figure 5 foods-12-01457-f005:**
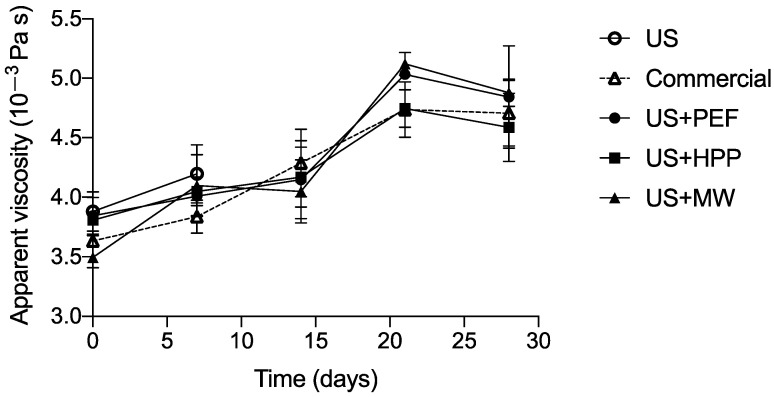
Apparent viscosity of milk samples during their shelf life. The values shown are the mean values (*n* = 4), and the standard error of means is shown as whiskers.

**Figure 6 foods-12-01457-f006:**
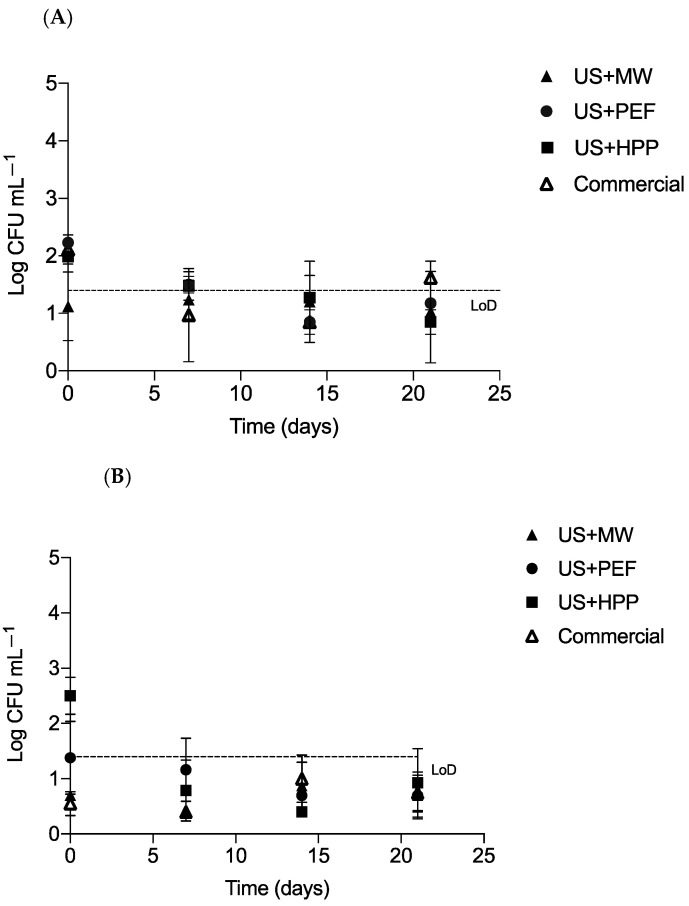
Microbial growth in milk during storage at 4 °C up to 28 days: (**A**) incubation at 37 °C; (**B**) incubation at 15 °C. Bacterial counts were measured on PCA plates (*n* = 6). The standard error of means is shown as whiskers. The detection level is indicated by the dotted line.

**Table 1 foods-12-01457-t001:** ζ-potential values of the fat globule emulsion of the different milks throughout the shelf life stored at 4 °C.

Days	0	7	14	21	28
Raw	−11.4 ± 0.8 ^aA^	−12.1 ± 0.3 ^aA^			
US	−17.5 ± 1.2 ^aC^	−20.1 ± 0.8 ^abB^	−19.1 ± 1.0 ^bB^		
Commercial	−14.6 ± 1.0 ^aB^	−14.7 ± 1.2 ^aB^	−14.4 ± 1.1 ^aA^	−14.6 ± 1.0 ^aA^	−15.3 ± 1.0 ^aA^
US + PEF	−18.5 ± 1.3 ^aCD^	−20.2 ± 0.8 ^aB^	−19.9 ± 0.7 ^aB^	−19.3 ± 1.2 ^aB^	−19.4 ± 0.7 ^aB^
US + MW	−20.1 ± 1.1 ^aD^	−20.2 ± 0.3 ^aB^	−19.4 ± 1.2 ^aB^	−18.9 ± 1.0 ^aB^	−18.5 ± 1.0 ^aB^
US + HPP	−19.7 ± 1.2 ^aD^	−20.5 ± 0.7 ^aB^	−19.7 ± 0.9 ^aB^	−18.7 ± 1.0 ^aB^	−19.2 ± 0.7 ^aB^

The same lowercase letter in the same row indicates that there are no significant differences (*p* = 0.05) for the same technology over the shelf life. The same capital letter in the same column indicates that, for the same day of shelf life, there are no significant differences (*p* = 0.05) between the technologies applied.

**Table 2 foods-12-01457-t002:** Values of consistency coefficient (*K*) and the flow behavior index (*n*) were obtained after fitting the flow curves to the Power Law model for each treated milk during the shelf life. For all of the results, the R^2^ ranged from 0.97 to 0.99.

Days	0	7	14	21	28
	K-index				
Raw	0.030 ± 0.007 ^A^				
US	0.024 ± 0.005 ^aAC^	0.058 ± 0.013 ^aA^	0.042 ± 0.013 ^aA^		
Commercial	0.046 ± 0.003 ^aB^	0.023 ± 0.000 ^aB^	0.028 ± 0.010 ^aA^	0.027 ± 0.009 ^aA^	0.019 ± 0.000 ^aA^
US + HPP	0.026 ± 0.003 ^aAC^	0.032 ± 0.010 ^aAB^	0.019 ± 0.001 ^aA^	0.038 ± 0.008 ^aA^	0.030 ± 0.010 ^aA^
US + MW	0.040 ± 0.009 ^aAB^	0.036 ± 0.019 ^aAB^	0.042 ± 0.016 ^aA^	0.048 ± 0.011 ^aA^	0.022 ± 0.001 ^aA^
US + PEF	0.028 ± 0.002 ^aAC^	0.034 ± 0.009 ^aAB^	0.040 ± 0.09 ^aA^	0.038 ± 0.008 ^aA^	0.034 ± 0.015 ^aA^
	n-value				
Raw	0.689 ± 0.036 ^A^				
US	0.732 ± 0.028 ^aA^	0.618 ± 0.029 ^abA^	0.670 ± 0.026 ^bA^		
Commercial	0.627 ± 0.021 ^aAB^	0.736 ± 0.009 ^bA^	0.713 ± 0.047 ^bA^	0.717 ± 0.037 ^bA^	0.758 ± 0.000 ^bA^
US + HPP	0.724 ± 0.018 ^aAC^	0.701 ± 0.047 ^aA^	0.773 ± 0.001 ^aA^	0.695 ± 0.037 ^aA^	0.719 ± 0.046 ^aA^
US + MW	0.643 ± 0.037 ^aA^	0.698 ± 0.076 ^aA^	0.661 ± 0.045 ^aA^	0.675 ± 0.035 ^aA^	0.735 ± 0.015 ^aA^
US + PEF	0.704 ± 0.002 ^aA^	0.690 ± 0.038 ^aA^	0.684 ± 0.031 ^aA^	0.702 ± 0.028 ^aA^	0.718 ± 0.048 ^aA^

The same lowercase letter in the same row indicates that there are no significant differences (*p* = 0.05) for the same technology over the shelf life. The same capital letter in the same column indicates that, for the same day of shelf life, there are no significant differences (*p* = 0.05) between the technologies applied.

## Data Availability

All related data and methods are presented in this paper. Additional inquiries should be addressed to the corresponding author.

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
