# Peer review of "Effect of the Application of Ultrasound to Homogenize Milk and the Subsequent Pasteurization by Pulsed Electric Field, High Hydrostatic Pressure, and Microwaves"

_foods, 2023, doi:10.3390/foods12071457_

Round 1

Reviewer 1 Report

Journal : Foods

Effect of the application of ultrasounds to homogenize milk and the subsequent pasteurization by pulsed electric fields, high hydrostatic pressure, and microwaves

Comment to authors.

2,3,4: Non-thermal techniques: Use their correct format i.e., ultrasound, pulsed electric field

28: Mention the correct unit of microbial count

31: Best pasteurization treatment with results should be mentioned

40. Please check these papers about non-thermal technologies and cite it. :

Combined Effect of Ultrasound and Microwave Power in Tangerine Juice Processing: Bioactive Compounds, Amino Acids, Minerals, and Pathogens. Processes10(10), 2100.

Applications of innovative non-thermal pulsed electric field technology in developing safer and healthier fruit juices. Molecules, 27(13), 4031.

Investigation of the effects of non-thermal, combined and thermal treatments on the physicochemical parameters of pomegranate (Punica granatum L.) juice. Food Science and Technology Research25(3), 341-350.

Pulsed electric field (PEF) of goat milk: Impact on Escherichia coli ATCC 8739 and vitamin constituents. Journal of Food Process Engineering, 44(9), e-13779.

49. This statement need citation. Please check this latest paper about ultrasound: Emerging trends for nonthermal decontamination of raw and processed meat: Ozonation, high-hydrostatic pressure and cold plasma. Foods, 11(15), 2173.

50-54. You have mention different studies, but you have mentioned one study only. Please add few more references.

57. Need reference.

63. Reference is not numbered.

67. Need reference. I am mentioning two review articles which are published on the impact of different nonthermal technologies on the milk and dairy products. You can read them and add some discussion from it. Food Science and Technology, 41(2): 279-294 &  International Journal of Dairy Technology, 72(4), 481-495.

121: Figure 1, Homogenization time is not mentioned

121: Figure 1, commercial homogenization and pasteurization conditions should be elaborated and HPP temperature mentioned should be cross checked

135: Use correct unit of energy density

144: Unit of temperature/ cross check temperature

147. PEF system

158. MW system

240-247. Please merge these paragraphs.

262,264: Proper labelling of figures

266: Figure 3A and 3B, check dependent and independent variables and their units as well

266: Figure 3B, use another format to distinguish among different treatments applied to milk

298: There is a need to mention novel or non-thermal technologies applied

314: Figure 4 should be labelled properly

331: Proper labelling of table is needed i.e., throughout storage of 28 days and stored at _C temperature

343: There is a need to mention the range of ζ –potential

366: Mention the correct non-thermal technology employed

509: Mention the best non thermal technique for pasteurization for future applications

The references should be updated.

Reviewer 2 Report

This paper reports a series of experiments on the impact of various treatments on homogenisation of milk and the stability of the subsequent emulsion.  The work is interesting (if the results are not all that surprising) and overall I am happy to recommend the paper, after the authors have considered the following suggestions.

1.     The work refers to pasteurisation by HPP and PEF, but I am not sure legally in Europe this term can be used for non-thermal treatments, as they have not been shown to achieve the specific pathogen elimination required for this term to be used (as acknowledged on line 465).  Thus, the implication of this in the title should be removed, and also throughout the paper (e.g., on line 404).

2.     The paper broadly ignores discussion of the impact of the treatments on milk proteins, and whether particle sizes and zeta potentials measured could show effects on caseins as well as fat globules.  In particular, the association of caseins with fat globules to stabilise the greatly increased surface area following homogenisation, which is well known, and the resulting impact on zeta-potential are not discussed (unlike association of denatured whey proteins, mentioned on line 355).  There is also reference to denaturation of caseins (line 414), which does not make dairy science sense – the whey proteins might be denatured, but the caseins will like either aggregate or dissociate under pressure, or perhaps form complexes with denatured whey proteins in heated milk. 

3.     The treatments (heating and homogenisation) applied to the commercial milk should be stated, and in a number of cases it would be more interesting to show the results for both the raw and conventionally treated milk.  In Fig. 3A it looks like this milk is not homogenised, which should be clarified.

4.     The photos take up a lot of space but do not really show a lot, in my view, and could be deleted.

5.     The increase in b-value (taken as indicating yellowness) could actually reflect protein or fat changes as the b-value can be very sensitive to protein aggregation processes such as rennet or acid coagulation.

6.     In the microbiology plots (Figure 7) it might be useful to include the untreated milk values somehow.  There is also in this section a lot of reference to literature to explain potential impact of the treatments, but some apparently quite significant data for inactivation of Listeria monocytogenes by PEF are referred to but not shown, which seems like it would have been worth including?

Some minor points are as follows:

1.     ‘ultrasounds’ in the title should be ‘ultrasound’

2.     Line 168: was there no holding time at 78C for the microwave-heated milk?

3.     The calculation of delta E values for Figure 5 is unclear – was the reference the commercial milk at day 0 or on each day of measurement?

4.     Line 419 refers to explaining the impact of treatments on viscosity of milk, but Figure 6 shows no effect?

Round 2

Reviewer 1 Report

Greatly revised.